# REPROMPTING: AUTOMATED CHAIN-OF-THOUGHT PROMPT INFERENCE THROUGH GIBBS SAMPLING

## ABSTRACT

We introduce Reprompting, an iterative sampling algorithm that searches for the Chain-of-Thought (CoT) recipes for a given task without human intervention. Through Gibbs sampling, we infer CoT recipes that work consistently well for a set of training samples. Our method iteratively samples new recipes using previously sampled solutions as parent prompts to solve other training problems. On five Big-Bench Hard tasks that require multi-step reasoning, Reprompting achieves consistently better performance than the zero-shot, few-shot, human-written CoT, Auto-CoT and self-consistency decoding baselines. Overall, Reprompting brings up to +17 point improvements over the previous state-of-the-art method that uses human-written CoT prompts.

## 1 INTRODUCTION

Few-shot prompting with large language models (LLMs) has revolutionized the landscape of natural language processing. Given natural language instructions and a few demonstrations as in-context examples, LLMs can quickly adapt to new tasks, approaching or even surpassing the performance of models fine-tuned on larger datasets on a wide range of tasks (Brown et al., 2020). However, such prompting techniques fall short on tasks that require multi-step reasoning and constraint propagation (Wei et al., 2022), such as *logical deduction* in the Big-Bench Hard benchmark (Suzgun et al., 2022). To address these limitations, prior works proposed to teach LLMs to reason step by step like humans by prompting them with chain-of-thought (CoT) reasoning steps for a few example problems (Wei et al., 2022). Despite the improved performance, such a method requires human experts with not only the task knowledge but also an understanding of how prompting works to craft the CoT prompt for each task (Zamfirescu-Pereira et al., 2023), which limits the scalability and generalizability of the method. Furthermore, a problem can be reasoned in many different ways, and some of them may work well on some LLMs but not on others. To fairly compare the performance of various LLMs on each task, we need to find the CoT prompt that works best for each model in a feasible way, which remains a challenge.

In this paper, we propose *Reprompting*, an iterative sampling algorithm that **automatically** finds effective CoT prompt for each model given a few question-answer pairs on a task without human intervention. Specifically, the algorithm aims to infer a set of CoT recipes that perform consistently well as few-shot in-context examples for solving the set of training problems. We frame it as a problem of sampling from a joint distribution of CoT recipes, which is infeasible to characterize directly but can be approached using Gibbs sampling – we initially sample the recipes by zero-shot prompting and then iteratively sample new recipes by concatenating a few old recipes as the prompt to solve a different training problem, which eventually converges to a set of recipes that share similar chains of thought for effectively solving the training problems. A handful of these CoT solutions from the training set then serve as effective CoT recipes for solving test problems.

We evaluate *Reprompting* on five Big-Bench Hard (BBH) tasks (Suzgun et al., 2022) that require multi-step reasoning, using ChatGPT (OpenAI, 2023) and InstructGPT (Ouyang et al., 2022) as LLMs. *Reprompting* consistently outperforms zero-shot, few-shot, human-written CoT, Auto-CoT (Zhang et al., 2022) and self-consistency decoding (Wang et al., 2022b) baselines. *Reprompting* also facilitates model combination by using different LLMs for initializing and sampling new recipes. Empirically, leveraging ChatGPT to sample initial recipes for InstructGPT brings up to +71 point improvements over using InstructGPT alone and even outperforms ChatGPT alone on certain tasks. Furthermore,

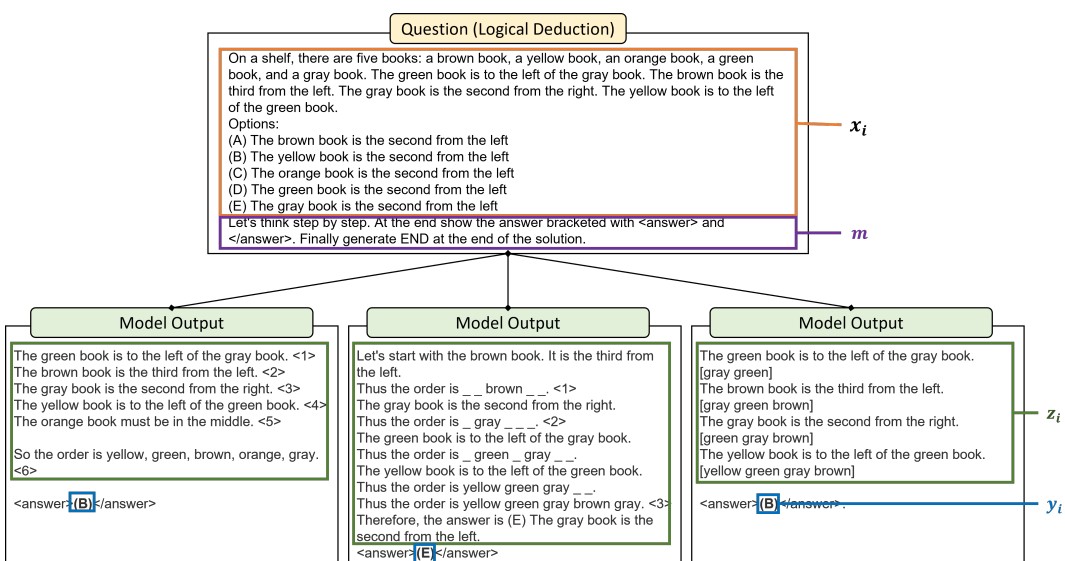

Figure 1: An example that ChatGPT can propose various different solutions to the same problem in zero-shot.

*Reprompting* outperforms the state-of-the-art, human-written CoT prompting on BBH tasks by up to +17 points. Lastly, our results confirm that the CoT recipes that work well on one model may work poorly on another, even when the latter may approach the best performance using prompts optimized for itself. These findings emphasize the need to optimize the CoT for each model for fairer comparisons.

## 2 *Reprompting*: PROMPT INFERENCE THROUGH GIBBS SAMPLING

### 2.1 STATISTICAL ASSUMPTIONS BEHIND IN-CONTEXT LEARNING

**In-context learning** has become the cornerstone of evaluating large language models (LLMs) (Brown et al., 2020; Srivastava et al., 2022). To facilitate this evaluation approach, data is provided for a large number of different tasks, with each task consisting of dozens or, more often, hundreds of instances with varying problem setup and question texts $x_i$ and their corresponding text answers $y_i$, where $i \in [1..N]$ and $N$ is the number of problem instances for the task. Statistically, in-context learning is based on the following assumption (which was verified empirically in Brown et al. (2020)):

$$p_{LLM}(y_j|\{x_i,y_i\}_{i\neq j},x_j) \approx p_{LLM}(y_j|\{x_i,y_i\}_{i\in S_j},x_j), \quad \forall S_j \subset [1,N]\setminus\{j\}, |S_j|=K \quad (1)$$

that is, the probability $p_{LLM}(y_j|\{x_i,y_i\}_{i\in S_j},x_j)$ of a given LLM generating the answer text $y_j$ when prompted with the concatenation of the few-shot examples $\{x_i,y_i\}_{i\in S_j\subset[1,N]\setminus\{j\}}$ and the test question $x_j$ (i.e. $[x_{S_j^1},y_{S_j^1},...,x_{S_j^K},y_{S_j^K},x_j]$) is close to the probability $p_{LLM}(y_j|\{x_i,y_i\}_{i\neq j},x_j)$ conditioned on all training examples $\{x_i,y_i\}_{i\neq j}$. In other words, any collection of $K$ examples of question-answer pairs $\{x_i,y_i\}_{i\in S_j}$ can be used as a prompt prior to the test question $x_j$ for an LLM to predict the answer $y_j$. In practice, however, the choice and even the order of examples can have a substantial impact on the test performance (Lu et al., 2022). And more importantly, the performance can be significantly enhanced by incorporating auxiliary knowledge or human-written instructions in a prompt (Shwartz et al., 2020; Zelikman et al., 2022; Nye et al., 2021), particularly in the form of Chain-of-Thought (CoT) reasoning (Wei et al., 2022; Wang et al., 2022b; Zhou et al., 2022; Creswell et al., 2022; Wang et al., 2022a; Liu et al., 2022; Kojima et al., 2022; Li et al., 2022).

In-context learning with CoT (Wei et al., 2022) can be seen in a similar light, statistically. In addition to the question-answer pairs $\{x_i,y_i\}$, the CoT prompt also contains worked out step-by-step reasoning "recipes" $z_i$ in text, which are inserted between the question and answer: $\{x_i,z_i,y_i\}$. These recipes can play two roles. First, they further explain the intent of the question $x_i$, as a small collection of question-answer pairs alone may be insufficient to disambiguate among different patterns an LLM

might detect. The second role is more important: it provides step-by-step guidance on one problem and thus teaches an LLM to solve similar problems following the same routine as it continues the text conditioned on the previous tokens. In the extreme, with prompts that strictly regiment self-attention, GPT models can be turned into Turing Machines to execute standard computer algorithms (Jojic et al., 2023). In practice, the CoT prompts commonly used in prior work fall somewhere between colloquial explanations and regimented recipes. Formally, **in-context learning with CoT** is based on the following assumption (which was empirically verified in Wei et al. (2022)):

$$p_{LLM}(z_j, y_j | \{x_i, z_i, y_i\}_{i \neq j}, x_j, m) \approx p_{LLM}(z_j, y_j | \{x_i, z_i, y_i\}_{i \in S_j}, x_j, m), \quad \forall S_j \subset [1,N] \setminus \{j\}, |S_j| = K \tag{2}$$

that is, the probability $p_{LLM}(z_j, y_j | \{x_i, z_i, y_i\}_{i \in S_j}, x_j, m)$ of a given LLM generating a step-by-step solution $z_j$ followed by the correct answer $y_j$ when prompted with the concatenation of the few-shot examples with CoT $\{x_i, z_i, y_i\}_{i \in S_j \subset [1,N] \setminus \{j\}}$, the test question $x_j$, and a special message $m$ (i.e. $[x_{S_j^1}, m, z_{S_j^1}, y_{S_j^1}, ..., x_{S_j^K}, m, z_{S_j^K}, y_{S_j^K}, x_j, m]$) is close to the probability $p_{LLM}(z_j, y_j | \{x_i, z_i, y_i\}_{i \neq j}, x_j, m)$ conditioned on all training solutions $\{x_i, z_i, y_i\}_{i \neq j}$. Here, the special message $m$ is an instruction text that is independent of specific questions and is appended to each question text. It can be task-specific or generic, as in the case of our experiments. Specifically, we use the following message for all tasks:

```
m = Let's think step by step. At the end, show your answer bracketed with
    <answer> and </answer>. Finally generate END at the end of the solution.
```

This message instructs the model to generate the step-by-step solution $z_j$ prior to the answer text $y_j$ and the specific format to present the answer.[1] In fact, such an instruction message can trigger instruction-tuned LLMs to generate step-by-step solutions given $[x_j, m]$ alone without any demonstration examples (i.e. $K = 0$), as illustrated in Figure 1. These solutions follow varying styles and often lead to incorrect answers. However, we argue that good recipes for solving the set of problems on a given task can evolve from these zero-shot solutions. In the next section, we introduce *Reprompting*, an iterative sampling algorithm that automatically produces the CoT recipes for a given set of problems without human intervention.

## 2.2 Prompt Inference Through Gibbs Sampling

We introduce the *Reprompting* algorithm, which aims to find a set of CoT recipes $z_i$ that work **consistently** well as few-shot in-context examples for a dataset $\{x_i, y_i\}_{i=1}^N$. Specifically, we formulate it as the problem of sampling from a joint distribution

$$p(z_1, z_2, ... z_N | \{x_i, y_i\}_{i=1}^N, m) \tag{3}$$

such that the conditional distributions $p(z_j | z_1, ..., z_{j-1}, z_{j+1}, ... z_N, \{x_i, y_i\}_{i=1}^N, m)$ satisfy the in-context learning assumption (2) that enables generalization with few-shot examples, i.e. the probability $p_{LLM}(z_j, y_j | \{x_i, z_i, y_i\}_{i \in S_j}, x_j, m)$ under the LLM is high and approximately invariant to the choice of examples $S_j$. Thus, we have the approximation

$$\begin{aligned}
&p(z_j | z_1, ..., z_{j-1}, z_{j+1}, ... z_N, \{x_i, y_i\}_{i=1}^N, m) \\
&\propto p_{LLM}(z_j, y_j | \{x_i, z_i, y_i\}_{i \neq j}, x_j, m) \\
&\approx p_{LLM}(z_j, y_j | \{x_i, z_i, y_i\}_{i \in S_j}, x_j, m), \quad \forall S_j \subset [1,N] \setminus \{j\}, |S_j| = K
\end{aligned} \tag{4}$$

Without characterizing the joint distribution, we can use Gibbs sampling (Geman & Geman, 1984) to generate such samples $\{z_1, z_2, ... z_N\}$ by first sampling $\{z_1, z_2, ... z_N\}$ independently from the distributions $p(z_j | x_j, y_j)$, and then iteratively drawing samples from the conditional distributions $p(z_j | z_1, ..., z_{j-1}, z_{j+1}, ... z_N, \{x_i, y_i\}_{i=1}^N, m)$. Based on the approximation (4), we can sample $z_j$ by randomly picking $K$ data points (excluding $j$) and then sampling $z_j$ with weights proportional to the conditional probability

$$p_{LLM}(z_j, y_j | \{x_i, z_i, y_i\}_{i \in S_j}, x_j, m) = p_{LLM}(z_j | \{x_i, z_i, y_i\}_{i \in S_j}, x_j, m) \cdot p_{LLM}(y_j | \{x_i, z_i, y_i\}_{i \in S_j}, x_j, m, z_j) \tag{5}$$

---

[1] This enables us to separate the generated answer $y_j$ from the step-by-step solution $z_j$ and forces the model to stop after generating the answer.

One way to approximate it is to sample several $\hat{z}_j$ from the LLM conditioned on the concatenation $[x_{S_j^1}, m, z_{S_j^1}, y_{S_j^1}, ..., x_{S_j^K}, m, z_{S_j^K}, y_{S_j^K}, x_j, m]$, compute the weight for each $\hat{z}_j$ using the model's probability of the correct answer $y_j$ conditioned on $[x_{S_j^1}, m, z_{S_j^1}, y_{S_j^1}, ..., x_{S_j^K}, m, z_{S_j^K}, y_{S_j^K}, x_j, m, \hat{z}_j]$, and sample a $z_j$ from $\{\hat{z}_j\}$ based on the weights. In practice, however, the model's probability of a given text may not be accessible. Thus, we approximately sample $z_j$ by sampling $\hat{z}_j$ and $\hat{y}_j$ from $p_{LLM}(z, y | \{x_i, z_i, y_i\}_{i \in S_j}, x_j, m)$ and then reject $\hat{z}_j$ with a probability of $p_{rej}$ if $\hat{y}_j \neq y_j$. Otherwise, we accept $\hat{z}_j$ and update the sample. Algorithm 1 shows the complete *Reprompting* algorithm consisting of the initialization and iterative sampling steps, which can be realized using different LLMs. Note that we set the rejection probability $p_{rej}$ in a way that allows solutions that lead to incorrect answers to be kept occasionally, as these solutions may still contain useful segments that evolve into good recipes through *Reprompting*.

---

**Algorithm 1:** *Reprompting* algorithm

**Input :** Training set $\{x_i, y_i\}_{i=1}^N$, number of shots $K$, number of iterations $M$, rejection
  probability $p_{rej}$, the initialization model $LLM_1$ and the sampling model $LLM_2$

1  **Initialization:**
2  **for** *each j* **do**
3  $\quad$ $z_j \leftarrow \emptyset$
4  $\quad$ Sample $\hat{z}_j, \hat{y}_j \sim p_{LLM_1}(z, y | x_j, m)$
5  $\quad$ Sample $u \sim Uniform([0, 1])$
6  $\quad$ **if** $\hat{y}_j = y_j$ *or* $u > p_{rej}$ **then**
7  $\quad\quad$ $z_j \leftarrow \hat{z}_j$
8  $\quad$ **end**
9  **end**
10  **Sampling:**
11  **repeat**
12  $\quad$ Randomly select $j \in [1, N]$
13  $\quad$ Randomly select $S_j \subset [1, N] \setminus \{j\}$ of size $K$
14  $\quad$ Sample $\hat{z}_j, \hat{y}_j \sim p_{LLM_2}(z, y | \{x_i, z_i, y_i\}_{i \in S_j}, x_j, m)$
15  $\quad$ Sample $u \sim Uniform([0, 1])$
16  $\quad$ **if** $\hat{y}_j = y_j$ *or* $u > p_{rej}$ **then**
17  $\quad\quad$ $z_j \leftarrow \hat{z}_j$
18  $\quad$ **end**
19  **until** *convergence or M iterations are reached*

---

Ideally, the algorithm should converge to the point where the probability $p_{LLM}(z_j, y_j | \{x_i, z_i, y_i\}_{i \in S_j}, x_j, m)$ is high and agnostic to the choice of $S_j$, which leads to a set of $\{z_j\}$ that work well as a prompt for solving similar problems in a separate test set.

The algorithm can also be viewed as a variant of evolutionary algorithms: 1) First, we generate the initial population of individuals (where each individual is a CoT recipe given a problem). 2) Next, we repeat the following regeneration steps iteratively: 2a) we first evaluate the fitness of each CoT recipe by comparing the answer that follows the recipe with the correct answer and weed out the least-fit recipes; 2b) we then breed new individuals through crossover and mutation by randomly selecting K recipes from the population as parent recipes, which are then used to prompt the LLM to generate recipes for a new problem. By repeating the 2a and 2b steps, initial recipes can be recombined (Figure 4) and evolve into better recipes (Figure 3) through iterations. And eventually, the fittest recipes (i.e. ones that lead to more accurate solutions) will survive.

During testing, we select $K$ tuples $\{x_i, z_i, y_i\}$ from the inferred $\{z_j\}$ based on the training accuracy when using each tuple individually in a prompt.

## 3 EXPERIMENTAL SETUP

We evaluate the *Reprompting* algorithm against various baselines including zero-shot, few-shot, Chain-of-Thought (CoT), Chain-of-Thought combined with self-consistency decoding (Wang et al., 2022b), and Auto-CoT (Zhang et al., 2022) on five challenging tasks in the Big-Bench Hard (BBH) benchmark (Suzgun et al., 2022): *Logical Deduction*, *Geometric Shapes*, *Object Counting*, *Penguins in a Table*, and *Temporal Sequences*. We choose these tasks from the BBH benchmark because these

are the tasks that require multi-step reasoning and have been shown to benefit substantially from human-written CoT in prior work (Suzgun et al., 2022).

***Reprompting***    For each task, we randomly select 20 training examples from the Big-Bench dataset excluding the test examples in the BBH benchmark.[2] We experiment with having $k \in \{1, 3\}$ clones of the same training example in the set $\{x_i, y_i\}_{i=1}^N$ to allow for more diverse recipe samples (so the number of recipes we need to sample from the joint distribution (3) is $N = 20 * k$) and choose $k$ that obtains the highest training accuracy. We set the number of shots by $K = 5$. We run *Reprompting* for a maximum of $M = 20,000$ iterations. We allow for early stopping if the average training accuracy stops increasing for $1,000$ iterations. For the rejection probability, we experiment with $p_{rej} \in \{0.95, 0.99\}$ and choose $p_{rej} = 0.99$ as it leads to higher training accuracy on various tasks.

**Baselines**    For **zero-shot prompting**, we only include the test question $x_i$ and the special message $m$ in the prompt, which triggers the model to generate a step-by-step solution prior to the answer text. For **few-shot prompting**, we randomly select 20 training examples in the same way as in *Reprompting* and concatenate these examples in the form of question-answer pairs in the prompt, followed by the test question. For **CoT prompting**, we use the human-written CoT prompts from Suzgun et al. (2022). For **CoT with self-consistency decoding**, we use the same CoT prompts and follow Wang et al. (2022b) by sampling 10 reasoning paths per question and taking the majority vote on the answer. For **Auto-CoT** (Zhang et al., 2022), since the original Auto-CoT algorithm differs from our setting as it focuses on the unsupervised setting without exploiting any labeled examples, we adapt the algorithm to our few-shot setting where it follows the original algorithm to generate diverse CoT recipes through zero-shot prompting but selects the demonstration examples based on the training accuracy (on the few-shot labeled examples) when used individually in a prompt.[3]

**Large Language Models (LLMs)**    We experiment with two powerful LLMs including Chat-GPT (gpt-3.5-turbo; OpenAI (2023)) and InstructGPT (text-davinci-003; Ouyang et al. (2022)). We also experiment with a combo model for *Reprompting* where we use ChatGPT as $LLM_1$ for initialization and InstructGPT as $LLM_2$ for sampling. For both LLMs, we set the maximum number of output tokens to 500, $TopP = 0.5$, and zero frequency or presence penalty. Additionally, we include "END" as the stop word. We set the temperature to 1.0 for *Reprompting* and 0.0 for testing.

**Evaluation Protocol**    We extract the final answer from the model output by extracting the text between "<answer>" and "</answer>", except for the CoT baseline where we extract the final answer in the same way as in Suzgun et al. (2022). We measure accuracy based on exact match by comparing the extracted answer with the ground truth.

## 4    RESULTS

Table 1 compares the performance of *Reprompting* with the previous state-of-the-art and the baseline prompting techniques. Repeating the experiments from Suzgun et al. (2022) with ChatGPT – using the same human-engineered CoT prompts – confirms the previous finding that few-shot in-context prompting improves the performance over zero-shot (Brown et al., 2020) and that CoT prompting outperforms both zero-shot and few-shot prompting by a large margin. Human-written CoT prompting requires costly prompt engineering, as not all CoT recipes work equally well on LLMs (Madaan & Yazdanbakhsh, 2022; Jojic et al., 2023). Crucially, we show that using *Reprompting*, LLMs can achieve better performance compared to the existing CoT prompts, but without requiring any human guidance on how to solve problems step by step. Specifically, comparing the performance of ChatGPT using *Reprompting* with ChatGPT using the best human-written CoT prompts from Suzgun et al. (2022), *Reprompting* achieves consistently higher scores on all tasks.

Additionally, we compare *Reprompting* with self-consistency (SC) (Wang et al., 2022b) decoding and the few-shot version of Auto-CoT (Zhang et al., 2022). First, CoT+SC improves over CoT on

---

[2]Except for *Penguins in a Table* where there are only three samples in the Big-Bench dataset that are excluded from BBH, so we randomly select 17 more examples from BBH into the training set.

[3]The original Auto-CoT algorithm selects the demonstration examples from the pool of zero-shot recipes based on the diversity of the demonstration questions.

| BBH Task | SOTA | ZS | FS | CoT ChatGPT | CoT+SC | AutoCoT | *Reprompting* ChatGPT | InsGPT | Chat+Ins |
|---|---|---|---|---|---|---|---|---|---|
| Logical | 60.4 | 35.1 | 46.4 | 63.1 | 62.7 | 53.2 | **66.3** | 53.7 | 60.0 |
| Geometric | 56.0 | 13.6 | 20.0 | 58.0 | 60.0 | 52.4 | **72.8** | 40.8 | 64.4 |
| ObjectCount | 93.2 | 52.4 | 46.8 | 95.6 | 95.2 | 88.8 | 97.2 | 42.8 | **99.6** |
| Penguins | 81.5 | 50.7 | 60.3 | 67.1 | 71.2 | **85.6** | **85.6** | 78.1 | 82.9 |
| Temporal | 96.8 | 38.4 | 41.2 | 66.8 | 66.8 | 80.8 | 93.2 | 28.4 | **99.2** |
| **Average** | 77.6 | 38.0 | 42.9 | 70.1 | 71.2 | 72.2 | **83.0** | 48.8 | 81.2 |

Table 1: Performance of several large language models (LLMs) using *Reprompting* versus the baseline prompting methods on Big-Bench Hard (BBH) tasks. *SOTA* refers to the state-of-the-art performance among Instruct-GPT (text-davinci-002; Ouyang et al. (2022)), Codex (Chen et al., 2021), and PaLM 540B (Chowdhery et al., 2022) using CoT prompting from Suzgun et al. (2022). We also compare *Reprompting* with ChatGPT using *ZS* (zero-shot), *FS* (few-shot), *CoT*, *CoT+SC* (CoT prompting combined with self-consistency decoding (Wang et al., 2022b)) and *AutoCoT* (the few-shot version of Auto-CoT (Zhang et al., 2022)). For *Reprompting*, we show the performance of various LLMs – including *ChatGPT* (gpt-3.5-turbo; OpenAI (2023)), *InstructGPT* (text-davinci-003), and *Chat+Instruct* (a combo version that uses ChatGPT for initialization and InstructGPT at sampling steps).

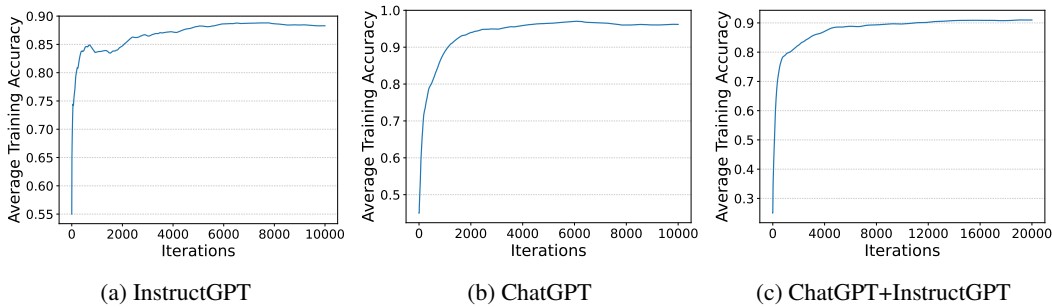

| (a) InstructGPT | (b) ChatGPT | (c) ChatGPT+InstructGPT |
|---|---|---|

Figure 2: Learning curves of the *Reprompting* algorithm using InstructGPT, ChatGPT, and the combo Chat-GPT+InstructGPT models on the *Logical Deduction* task. The y-axis shows the accuracy on training samples averaged over the current and all previous iterations.

two of the five tasks, but the improvements are not consistent. *Reprompting* outperforms CoT+SC by 2–26 points on all five tasks. When compared with Auto-CoT, *Reprompting* achieves +11 higher scores on average.

As expected, the performance of *Reprompting* depends on the LLM, and InstructGPT underperforms ChatGPT on most tasks. However, we show that by using ChatGPT just as **the initialization model** $LLM_1$ to bootstrap InstructGPT as $LLM_2$ in prompt optimization through *Reprompting*, we can improve performance over InstructGPT alone by 5–71 points and achieve competitive or even better performance than ChatGPT alone on four of the five tasks. We show in the Appendix why that is: while InstructGPT can follow a given recipe and even be used for recombining and evolving them, it cannot create useful and diverse initial solutions in a zero-shot manner. However, through *Reprompting*, we can use ChatGPT to "teach" InstructGPT the basic strategies for solving the training problems, which are then recombined and evolved by InstructGPT into better CoT prompts for InstructGPT.

Finally, our results demonstrate that *Reprompting* achieves up to +17 point improvement over the previous state-of-the-art results on BBH tasks using human-written CoT prompts (Suzgun et al., 2022). These findings highlight the potential of *Reprompting* as a powerful method for automating CoT prompting and combining the strengths of different LLMs to achieve better performance on a wide range of tasks.

***Reprompting* improves CoT recipes over iterations.** In Figure 2 the average training accuracy (averaged over iterations up to the current iteration) of the *Logical Deduction* task is plotted over the training iterations. For all three model variants, the initial training accuracy is relatively low, but it gradually increases (with occasional fluctuations) over iterations until convergence. This is the

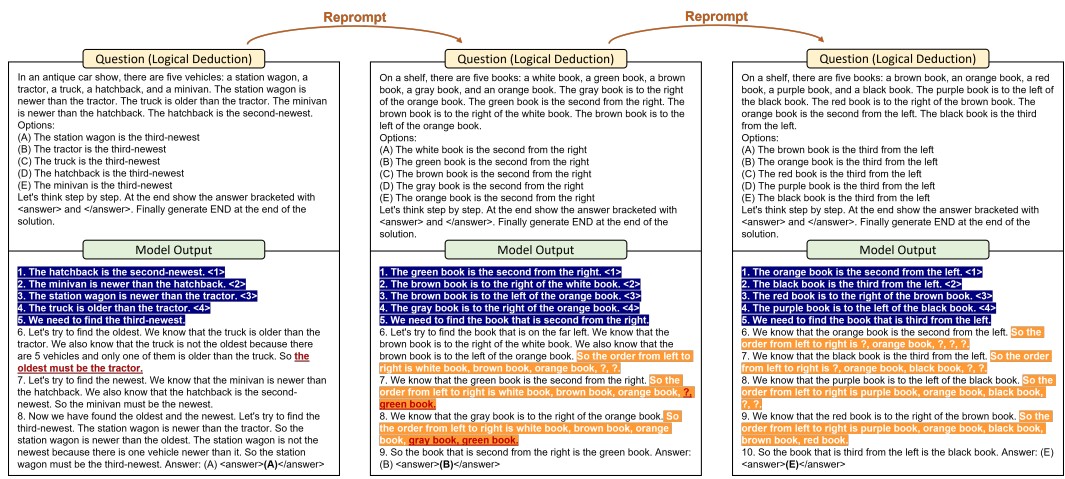

Figure 3: An example of how the CoT recipes evolve through *Reprompting*. In the left-most recipe, the model (ChatGPT) first reorders the constraints so that the ones with absolute ranking positions are considered prior to the ones with relative positions (highlighted in dark blue). Next, the model attempts to deduce the objects at specific positions but makes a mistake (see the red underlined part). Despite the error, this recipe still provides a useful strategy for solving similar problems: when using this recipe in the prompt to solve another problem, the model first adopts the same strategy to reorder the constraints and then proposes another way to deal with the constraints (highlighted in orange). Although the resulting solution still contains errors, it makes a good recipe for solving this type of problem. And thus, when using it as the new prompt to solve yet another problem, the model is able to follow the same recipe and deduce the correct answer.

result of evolution and recombination of the recipes associated with training examples (which was the motivation for the name *Reprompting*).

We observe that **even model outputs containing errors and unreasonable deductions can evolve into a high-quality recipe through *Reprompting*.** This is illustrated by the *Logical Deduction* example in Figure 3, when $K = 1$, where the model initially generates a recipe that is erroneous and contains illogical deductions. However, when this recipe is used as the new prompt for solving a similar problem, the model is able to exploit the useful components of the recipe and propose an alternative way to continue reasoning. Although the subsequent recipe still contains errors, it aids the model in correctly solving other problems when incorporated into a prompt. As a result, such recipes will be populated on other training samples, while the recipes that lead to low accuracy will eventually die out.

***Reprompting* combines fragments from different recipes into a better one.** *Reprompting* benefits from having multiple examples in the prompt, which allows the model to integrate various segments from different prompt recipes into a new recipe. As illustrated by the *Object Counting* examples in Figure 4, the model can combine large segments of reasoning steps, as well as small segments that address distinct cases to solve a more complex problem. The resulting prompts sometimes, but not always, share similarities with the human-written prompts (See the Appendix).

**Do the generated CoT recipes generalize across models?** We test the best-performing CoT recipes sampled from InstructGPT, ChatGPT, or InstructGPT+ChatGPT through *Reprompting* on the test set with both InstructGPT and ChatGPT. The results (Table 2) indicate that the CoT recipes optimized for one model may not work as well for other models. Specifically, we observe that on tasks such as *Logical Deduction* and *Object Counting*, the best CoT recipes achieve similar performance on both InstructGPT and ChatGPT. However, on *Geometric Shapes*, *Penguins in a Table* and *Temporal Sequences*, the best CoT prompts optimized for $LLM_2$ work well on $LLM_2$, but poorly with the other

| Tasks | InsGPT | ChatGPT |
|---|---|---|
| Logical | 65.9 | **66.3**[*] |
| Geom. | 53.6 | **72.8**[*] |
| Objects | **99.6**[*] | 96.8 |
| Penguins | **85.6**[*] | 76.7 |
| Temporal | **99.2**[*] | 81.6 |

Table 2: Performance of LLMs using the same best performing CoT recipe for each task from Table 1. The superscript [*] denotes that the recipe was originally reprompted for that LLM.

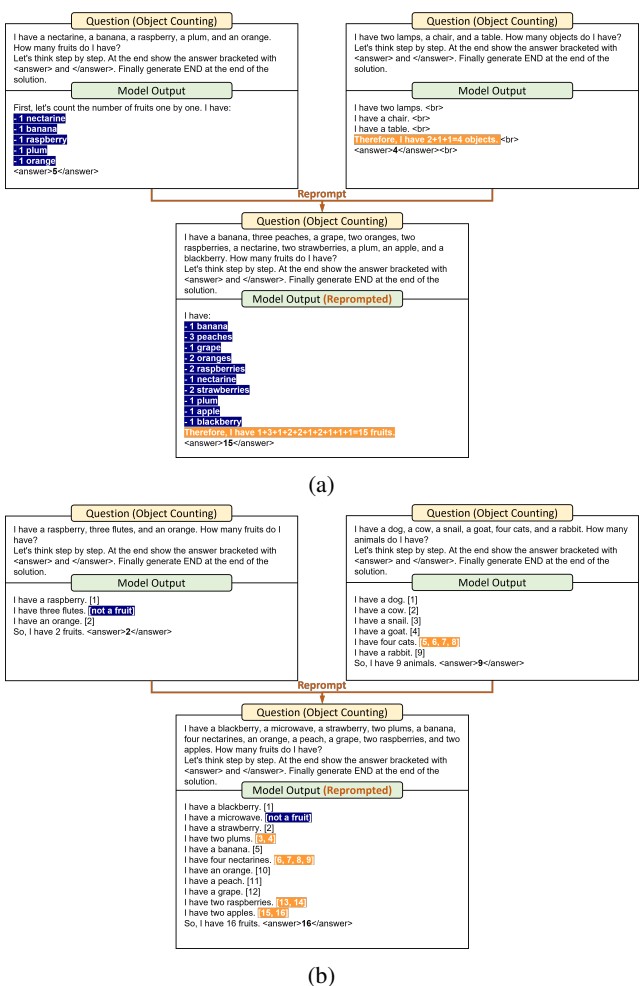

Figure 4: Examples of how fragments from different recipes in a prompt can be (re)combined into a better recipe to solve a new problem through *Reprompting*.

LLM – using them on the other LLM leads to 9–19 points lower performance than testing with $LLM_2$ (see examples in Figure D.2). However, when using the prompts optimized for the testing LLM, it achieves 1–12 higher scores than those using the prompts optimized for the other LLM and sometimes even approaches the best performance among InstructGPT and ChatGPT. These results suggest that to make a fair comparison between different LLMs, one needs to optimize the CoT prompt for each model.

**Compute and Resources** We use the OpenAI APIs for all our experiments.[4] Running *Reprompting* costs around $80 (in US dollars) on gpt-3.5-turbo and $800 on text-davinci-003 based on the standard pricing,[5] while being exempted from any human cost. By contrast, CoT prompting requires manual prompt construction and engineering, which costs not only human labor (including the cost for humans to get familiar with the task itself and how LLM prompting works, write down various CoT solutions for each problem, test and optimize the solutions on the LLM) but also LLM queries, but these costs are typically neglected in previous works. In addition, previous works typically compare different LLMs using the same CoT prompt. While this strategy avoids additional costs for custimizing CoT prompt for each LLM (and with *Reprompting*, one can also save the cost by running it with ChatGPT and using the inferred CoT prompt on other LLMs), it risks making unfair comparisons as we have shown in Table 2 that the CoT prompt that works well on one model may be sub-optimal for another.

---

[4] https://platform.openai.com/docs/api-reference?lang=python
[5] https://openai.com/pricing

## 5 RELATED WORK

**In-Context Learning** is an emergent ability of LLMs as they scale up in model sizes and training data, where an LLMs can learn to perform a task from a few examples in the context (which is also referred to as few-shot prompting) (Brown et al., 2020). It has been shown to achieve promising few-shot and even zero-shot performance on various natural language processing (Brown et al., 2020; Schick & Schütze, 2020; Perez et al., 2021) and program synthesis (Austin et al., 2021) tasks.

**Reasoning via Chain-of-Thought Prompting** Chain-of-Thought (CoT) prompting is a technique that enables LLMs to perform complex reasoning tasks by prompting them with a few examples with step-by-step solutions (Wei et al., 2022; Suzgun et al., 2022). CoT prompting has been shown to improve performance on various reasoning tasks, such as arithmetic reasoning (Wei et al., 2022; Zhou et al., 2022), symbolic reasoning (Wei et al., 2022; Zhou et al., 2022), multi-hop question answering (Press et al., 2022; Arora et al., 2022), and natural language inference (Wang et al., 2022b). However, designing effective CoT prompts requires human experts with an understanding of both the task and the prompting technique (Zamfirescu-Pereira et al., 2023), which limits the scalability and generalizability of CoT prompting.

Several works have attempted to **automate the process of CoT prompt discovery**. Zhang et al. (2022) proposed Auto-CoT, which uses LLMs to generate CoT solutions for diverse training questions in zero-shot and integrates the generated CoT solutions in the prompt for solving test questions. This method differs from *Reprompting* in that: 1) it focuses on the unsupervised setting and exploits a large set of example questions without annotated answers, and 2) it relies more heavily on the correctness of the zero-shot recipes as it does not have any iterative algorithm (as in *Reprompting*) to further improve the recipes. In our experiments, we adapted Auto-CoT to the few-shot setting and showed that *Reprompting* outperforms the few-shot version of Auto-CoT.

Deng et al. (2022); Zhang et al. (2023) proposed to train an additional policy model to find the best prompt through reinforcement learning, but their approaches are limited to prompt optimization within a relatively small search space (i.e. it is restricted to the prompts that are either extremely short or within a small edit distance from an initial prompt). Zhou et al. (2023) proposed a method for automatically generating, scoring and selecting effective instruction messages *m* for zero-shot chain-of-thought reasoning, which is orthogonal and can be potentially combined with our algorithm. Paranjape et al. (2023) introduced a framework that automatically retrieves demonstrations of related tasks from a task library and generates CoT solutions for the new task. However, this framework still requires collective human efforts to write demonstrations for a diverse set of tasks in the task library. In contrast, our *Reprompting* algorithm enables LLMs to solve complex reasoning tasks without any human guidance. Additionally, Yoran et al. (2023) proposed a multi-chain reasoning (MCR) method that prompts LLMs to combine pieces of information from multiple chains of thought to predict the final answer, which differs from our method in two ways: first, MCR combines multiple CoT solutions to the same question at test time, while *Reprompting* combines CoT solutions generated for different training questions before testing; second, MCR combines solutions only once, whereas *Reprompting* iteratively samples new solutions and recombines them. As a result, *Reprompting* generates effective CoT recipes from only a few training examples, resulting in improved test performance without slowing down test inference.

## 6 CONCLUSION

We introduce *Reprompting*, an automated prompt inference algorithm which, without human effort, discovers effective chain-of-thought (CoT) prompts for each task given a few question-answer pairs. On five Big-Bench Hard (BBH) tasks, prompts discovered with *Reprompting* consistently outperform zero-shot, few-shot, human-written CoT, Auto-CoT (Zhang et al., 2022) and self-consistency decoding (Wang et al., 2022b) baselines. Furthermore, *Reprompting* facilitates model combination, which may significantly improve the performance of a weak LLM by using a stronger LLM to generate initial CoT solutions and then reprompting using the weak LLM to optimize the prompt for itself. Overall, *Reprompting* achieves up to +17 point improvements over the previous state-of-the-art on BBH tasks, which was based on human-written prompts. Our results suggest that LLM comparisons can be highly sensitive to the choice of CoT prompts, further emphasizing the need for automatic prompt discovery and optimization using algorithms such as *Reprompting*.

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
