# OpenReview forum: "Reprompting: Automated Chain-of-Thought Prompt Inference Through Gibbs Sampling"
_ICLR.cc/2024/Conference — Submitted to ICLR 2024_

### Official Review · Reviewer_5tZK · 2023-10-16

**Soundness:** 2 fair
**Presentation:** 2 fair
**Contribution:** 2 fair
**Rating:** 5
**Confidence:** 4

**Summary:**

The paper introduces Reprompting, an algorithm automatically seeks CoT recipes for a given task.
Reprompting iteratively samples new recipes using previously successful solutions as starting points. Experiments conducted on BBH tasks show that (i) Reprompting can transfer knowledge from GPT-3.5-turbo to text-davinci-003; (ii) Reprompting performs better than previous methods like CoT, AutoCoT, Self-Consistency.

**Strengths:**

- experimental results on BBH tasks show that the proposed Reprompting performs better than the SOTA CoT+SC and AutoCoT
- Reprompting can generate the cot receipts without expert knowledge
- the paper is easy to follow

**Weaknesses:**

- major: the algorithm is designed based on assumptions (1) and (2), which are **not standard** in LLMs.
- major: in experiments, is message $m$ (i.e., *Let’s think step by step. At the end, show your answer bracketed with <answer> and </answer>. Finally generate END at the end of the solution*) applied to all other baselines? If not, an ablation study is required to justify where the improvement comes from, i.e., the message or the proposed reprompting technique.
- major: experiments are weak: popular benchmarks (e.g., datasets used in Tables 2 and 3 of Self-Consistency) have not been used in the paper.
- major: justification for the contribution "*Reprompting can transfer knowledge from a stronger model to a weaker model" is not strong; only one model is studied, i.e.,  from GPT-3.5-turbo to text-davinci-003
- minor: no ablation study for the rejection probability (i.e., testing accuracy w.r.t. rejection probability)
- major: ComplexCoT (Complexity-Based Prompting for Multi-Step Reasoning) is also an important baseline but not compared with, and also combining it with Self-Consistency

**Questions:**

- Is the rejection sampling used in Sec 2.2 equivalent to drawing samples according to (5)?
- "*Note that we set the rejection probability $p_{rej}$ in a way that allows solutions that lead to incorrect answers to be kept occasionally, as these solutions may still contain useful segments that evolve into good recipes through Reprompting*", any examples? How many incorrect answers are in the generated CoT receipts?
- "*in-context learning is based on the following assumption: ...*", any credits on this assumption?
- "*Formally, in-context learning with CoT is based on the following assumption: ... *" any references?
- is reprompting sensitive to the choice of $\\{x_i, y_i\\}_{i=1}^N$?
- Will the proposed Reprompting be used in recent verification-based methods? e.g.
  - Large Language Models are Better Reasoners with Self-Verification
  - LEVER: Learning to Verify Language-to-Code Generation with Execution
  - Forward-Backward Reasoning in Large Language Models for Mathematical Verification
  - Making Language Models Better Reasoners with Step-Aware Verifier


## ----- Post-Rebuttal -----
Thanks for the authors' reply, which addressed some concerns about experiments. Thus, I increased my rating to 5.
The main reason for rejection is the assumptions required by the proposed algorithm are strong and difficult to check when applying the algorithm in practice

- the authors provided some references for the strong assumptions (1) and (2); however, it is still not convincing.
- based on the new results provided in tables 3/4 and results provided by https://arxiv.org/abs/2304.09797, Reprompting is worse than ComplexCoT (which also uses "let's think step by step") on some tasks by a large margin, I listed some numbers for comparison bellow:
& GSM8K & Number Theory
Complex CoT &  82.8 & 33.4
Reprompting &  79.5 & 28.5

Therefore, users need to check whether the assumptions are satisfied before applying the proposed algorithm.

> Complex-CoT relies on a pool of human-written CoT recipes to select from while Reprompting does not.
- this is a valid motivation, but I think writing eight CoT demos is easy.

---

> ### Author Response · Authors · 2023-11-22
>
> We thank the reviewer for the valuable comments and insightful questions. Here are some responses to the weaknesses and questions:
>
> - The assumptions in Eq(1)(2) are based on the empirical findings in [1,2] on a wide range of tasks. Specifically, [1] shows that the test performance does not vary much when increasing the number of supervised examples in the prompt from 8 to 32 or more. Chain-of-Thought (CoT) prompting further reduces the number of demonstrations needed -- the test performance remains similar when there is a varying number (but typically >= 3) of few-shot exemplars. These findings can be expressed formerly in mathematical forms as in Eq(1)(2), and have served as the base of LLM evaluation through few-shot prompting in prior work [3,4] . We have updated Section 2.1 to better explain it.
> -	The message “Let’s think step by step…” is applied to all the baselines, except for the few-shot baseline where the demonstrations only include the concatenated question answer pairs without intermediate steps, thus we didn’t include the “step-by-step” instruction in the message.
> -	We have added more experiments on 10 additional tasks and popular benchmarks, including 7 tasks from BigBench Hard (BBH), GSM8K (math word problems) and two from Hendrycks’ MATH (see Table 3 in Appendix). (Many tasks evaluated in self-consistency paper predate GPT3.5 and thus may have data contamination issue. The BBH data contains more recent and more challenging benchmark tasks, e.g. Date Understanding is considered more challenging than StrategyQA [3].) Results show that Reprompting still outperforms zero-shot and few-shot prompting consistently and substantially by 14-17 points on average. Compared with human-written CoT, Reprompting achieves better performance on all these tasks except on Date Understanding, Reasoning About Colored Objects, and MATH Algebra, on which the score differences are very small (<4 point). On average, Reprompting still outperforms CoT by +12.4 point on these tasks.
> -	We have downplayed the claim for the contribution "Reprompting can transfer knowledge from a stronger model to a weaker model" in the updated paper, as we observed it only on GPT3.5 to text-davinci-003. However, the finding is still interesting and serves as a starting point for further study in future works.
> -	We further conducted an ablation study on the rejection sampling and recombination process. Results in Table 5 (in Appendix) show that, without rejection sampling, the test performance degrades greatly. Always rejecting solutions that lead to wrong answers also leads to degradation. Additionally, not allowing multiple solutions to be recombined when sampling new solutions at the iterative sampling stage also hurts performance.
> -	We have conducted additional experiments to compare Reprompting with ComplexCoT (Table 4 in Appendix). On the three commonsense and arithmetic reasoning tasks, our method outperforms the baseline by +5.3 points on average. Note that Complex-CoT relies on a pool of human-written CoT recipes to select from while Reprompting does not. We are still running ComplexCoT+SelfConsistency (which is 8.5x more costly than Reprompting on GSM8K), but we don’t expect it to outperform Reprompting since SelfConsistency improves over CoT by <= 4 point in our previous experiment.
> -	Rejection sampling is used to simulate the sampling from Eq(5), but is not equivalent to it.
> -	We showed an example of incorrect solutions evolving into a correct one through iterations in Figure 3. The percentage of incorrect solutions kept in the initialization stage is in Figure 2 (at 0-th iteration), which is around 50-70%.
> -	The performance does not vary much given different choices of the prompt examples after running Reprompting, which is probably because the solutions learned through Reprompting follow the same reasoning paths.
> -	Yes, applying Reprompting to optimize the prompt for verification would indeed be a great future direction.
>
> [1] Language Models are Few-Shot Learners. Brown et al., NeurIPS 2020.
>
> [2] Chain-of-thought prompting elicits reasoning in large language models. Wei et al., NeurIPS 2022.
>
> [3] Challenging BIG-Bench tasks and whether chain-of-thought can solve them. Suzgun et al., 2022.
>
> [4] An Empirical Study on Challenging Math Problem Solving with GPT-4. Wu et al., 2023.

---

### Official Review · Reviewer_78Bd · 2023-11-01

**Soundness:** 3 good
**Presentation:** 3 good
**Contribution:** 3 good
**Rating:** 6
**Confidence:** 4

**Summary:**

This paper proposes reprompting, an iterative algorithm that continually improve chain-of-thought (CoT) reasoning from the LLMs' own outputs. The idea is that good in-context learning (ICL) demonstrations should work consistently well for all examples rather than being sensitive to ordering, etc. This problem is formulated as a sampling problem from the joint distribution and the authors propose to sample from it via Gibbs sampling. The authors then demonstrate the effectiveness of the proposed algorithm in ChatGPT model with several Big Bench Hard problems and show outperformance over standard zero-shot, few-shot and CoT prompting baselines.

**Strengths:**

- The approach is fairly novel, well-motivated and theoretically grounded: I think the formulation of finding good in-context learning examples as approximate sampling from an intractable joint probability distribution and using Gibbs sampling, in this case, are both well-motivated. I am convinced that this is a solid point for LLM prompting, a field of study on which the model is often treated as a black box, and the approaches proposed are often largely heuristic.
- The results seem promising, and given that improving LLM reasoning is an important problem, I think the method presented has clear potential in terms of significance.
- The paper is well-written overall.
- I reviewed a previous version of the manuscript; one of my major concerns was the lack of comparison to self-consistency (which improves over zero-shot and few-shot CoT with more computational resources, which makes the baselines more comparable to the proposed method, which is expensive) and baselines like AutoCoT and COSP (the authors added AutoCoT but not COSP; I encourage the authors to include discussions of all relevant work if possible); the authors added discussions and empirical comparisons to these, which significantly strengthened the empirical rigour of the paper. Similarly, I was concerned about the cost of the algorithm, but in this updated version, I can see that the authors have dedicated a section addressing these concerns and included an analysis of the transferability of the discovered in-context examples to justify their design choices. These have again made the paper much more convincing in terms of motivation.

**Weaknesses:**

- I am still a bit unconvinced that only five subtasks out of the 23 tasks of BIG-Bench Hard were considered, given that each subtask is quite small and choosing 5 out of 23 leads to doubts on whether the results are cherry-picked. In the previous reviewing round, the authors responded that they chose tasks where 1) CoT prompting is likely effective and 2) diverse, and they endeavoured to include additional tasks. However, I think in a practical setting, it's unlikely that practitioners will know beforehand whether CoT prompting is effective or not, and it is still interesting to see how methods such as Reprompting work *even when CoT prompting does not improve much*, this gives readers stronger faith that in the worst case scenario, the method at least does not deteriorate performance. Secondly, given the time allowance between the review cycles, I'd expect experiments on additional subtasks, but this is not present in the manuscript. To start with, I see this being the largest room for improvement of the current manuscript.

-- **Post-rebuttal** --

I thank the authors for their detailed feedback, which mostly addressed my concerns. I will stick to my original rating recommending acceptance, although some of the concerns raised by other reviewers should be addressed, including the discussions on the validity of the assumptions and potential comparison to additional baselines. Since ICLR allows the authors to revise the paper directly, I also think the authors should directly incorporate some of the responses and promised changes, such as the additional discussions on related works (I also think the additional experiments should be in the main text directly instead of in the appendix). A final remark is that the authors, in multiple responses, quoted the fact that methods like COSP do not outperform manual CoT as a justification; however, these methods are *zero-shot* whereas manual CoT and Reprompting are not, so these arguments should be carefully qualified.

**Questions:**

Please address the Weaknesses mentioned above.

---

> ### Author Response · Authors · 2023-11-22
>
> We thank the reviewer for the valuable comments and insightful questions.
>
> Due to improved compute access recently, we were able to run more experiments on 10 additional tasks, including 7 tasks from BigBench Hard (BBH), GSM8K (math word problems) and two from Hendrycks’ MATH (see Table 3 in Appendix). On BBH, we intentionally selected the subtasks on which CoT does not improve much or does not improve consistently, such as Formal Fallacies, Movie Recommendation, Ruin Names, Salient Translation Error Detection, and Word Sorting. Results show that Reprompting still outperforms zero-shot and few-shot prompting consistently and substantially by 14-17 points on average. Compared with human-written CoT, Reprompting achieves better performance on all these tasks except on Date Understanding, Reasoning About Colored Objects, and MATH Algebra, on which the score differences are very small (<4 point). On average, Reprompting still outperforms CoT by +12.4 point on these tasks.
>
> And interestingly, on tasks where CoT even underperforms zero-shot prompting, such as Movie Recommendation, Salient Translation Error Detection, and Word Sorting, Reprompting still improves over zero-shot prompting substantially. This suggests that not all CoT recipes improve model performance, and some may even lead to degradation. Reprompting serves as an effective algorithm for discovering and optimizing the CoT prompt to best exploit and compare LLMs.

---

### Official Review · Reviewer_kqkB · 2023-11-01

**Soundness:** 3 good
**Presentation:** 2 fair
**Contribution:** 2 fair
**Rating:** 5
**Confidence:** 4

**Summary:**

This paper proposes “Reprompting” a method to automatically generate chain-of-thought (CoT) prompts for large language models (LLMs). The hope with Reprompting is to alleviate the issue of manually handcrafting chain-of-thought prompts while retaining high performance. In a nutshell, Reprompting is inspired from Gibbs sampling and proceeds as follows:

1. Sample initial CoT $z_j$ and answer $y_j$ from an LLM1, for each query $x_j$ in a given support set.
2. Update $z_j$ and $y_j$ by sampling them from LLM2 conditioned on the query $x_j$, and the other CoT $z_{i \neq j}$ and answers $y_{i \neq j}$ in the support set.
3. Repeat step 2 until convergence.

The motivation for using two LLMs (LLM1 and LLM2) is to ensure diversity in the initial set of CoT prompts.
The authors demonstrate the effectiveness of Reprompting on 5 reasoning tasks from Big Bench Hard (BBH) and compare against human and automatic baselines for CoT prompting.

**Strengths:**

This paper has a couple of strong selling points, most notably its effectiveness given how easy it is to implement.

- Reprompting is simple enough to be widely adopted, and I feel confident I could reimplement it without trouble. It also makes minimal assumptions on the infrastructure needed to run the method, and is readily implementable with commercial APIs (the authors use OpenAI’s).
- Reprompting solves a real-world problem: designing chain-of-thought prompts can be quite costly if human-crafted, especially in a large-scale setting. Here the authors report that Reprompting costed between $80 and $800 USD, which is likely to be significantly less than the cost of hiring and training human annotators, and verifying their solution.
- Reprompting gets excellent results on the 5 BBH tasks it was tested on (with caveats below).

**Weaknesses:**

Here are some limitations of the work, roughly in order of importance.

- Missing baselines and ablations:
    - Parameter-efficient fine-tuning: since Reprompting is mainly pitched as a way to find high-performing prompts, it should also be compared against parameter-efficient methods such as soft-prompt tuning and LoRA. I’d be interested in both absolute performance (Table 1) and convergence rate (Figure 2).
    - Sampling ablation: at the top of page 4, the authors mention that the LLM likelihood is not always accessible so they approximate it by randomly rejecting new samples. First, this approximation needs more justification for why it’s appropriate. Second, I’d like to see quantitative evidence supporting this approximation — how much better does the method work when actually computing the sampling probabilities? While they may not be available to many practitioners, scientifically we ought to know how much we’re leaving on the table with the approximation.
    - Rewriting ablation: line 14 of Algorithm 1 doesn’t condition on the current CoT $z_j$, so the new one is sampled without reading prior guesses. How much better would Reprompting work if it had access to prior CoT guesses? This further breaks away from the original Gibbs sampling motivation but may give substantial improvements, according to Yang et al., 2023 (https://arxiv.org/abs/2309.03409).
- Limited benchmarks: since this paper rests on experimental evaluations (no theory), I’d expect it to be more thorough. In principle Reprompting could be used on any task and I was expecting results and Hendrycks Math, MMLU, and more.
- Exposition: the method feels like it’s been shoehorned to fit the Gibbs sampling narrative, with many approximations along the road. I’m not sure why that is the case since the authors don’t use this motivation later on. Similarly, the evolutionary algorithm view is briefly mentioned before Section 3, but why?

**Questions:**

See my weaknesses, but generally:

- How does Reprompting compare to parameter-efficient tuning methods?
- Can you provide a justification for the approximation at the top of p. 4?
- How much is left on the table due to this approximation? Is it an issue that generative LLMs are bad at estimating the likelihood of their samples? How does that affect the Gibbs sampling motivation?
- In terms of wall-clock time, how long did it take to converge in Fig. 2?
- The motivation for decoupling initialization and sampling is to improve the diversity of the initial prompts. Do you have evidence for this claim? From Table 1 it’s unclear when we should use 1 LLM for both vs decoupling them.
- Did you see convergence in the type of CoT prompts? Do they all end up following a similar format and is that problematic for some tasks?

---

> ### Author Response · Authors · 2023-11-22
>
> We thank the reviewer for the valuable comments and insightful questions. Here are some responses to the weaknesses and questions:
>
> - Compared with prompt tuning and fine-tuning approaches, the number of labeled samples required by our approach is much less. Prompt tuning and parameter-efficient fine-tuning approaches require 100 to several thousand labeled samples for training and validation [1,2,3,4,5], which is not suitable for the few-shot setting that we focus on in this paper (e.g. on BigBenchHard, some tasks only contain a few dozen labeled samples apart from the 250 test samples). By contrast, our approach only requires 20 labeled samples, and thus can be tested on these few-shot benchmarks. However, we did add an additional baseline that optimizes the CoT prompt based on human-written solutions (so it’s human effort + optimization) [6]. The results are presented in Table 4 in the updated Appendix. On the three commonsense and arithmetic reasoning tasks, our method outperforms the baseline by +5 points on average.
> -	Sampling ablation: Since the model likelihood is only available for text-davinci-003 (which is 10x more expensive than GPT3.5) but not GPT3.5, and sampling a set of candidates for reweighted sampling at each iteration is time-consuming, it takes much more time to run. But we are still trying to run it and incorporate the results in the updated version.
> -	Reprompting+Rewriting: This would indeed be an interesting way to further improve the performance of Reprompting, although it remains an open question how to instruct a model so that they can use recipes in a prompt differently (i.e. it should try to recombine the reasoning paths provided by recipes on other questions while improving (but not copying) the previous recipe on the same question).
> -	We have added more experiments on 10 additional tasks and popular benchmarks, including 7 tasks from BigBench Hard (BBH), GSM8K (math word problems) and two from Hendrycks’ MATH (see Table 3 in Appendix). Results show that Reprompting still outperforms zero-shot and few-shot prompting consistently and substantially by 14-17 points on average. Compared with human-written CoT, Reprompting achieves better performance on all these tasks except on Date Understanding, Reasoning About Colored Objects, and MATH Algebra, on which the score differences are very small (<4 point). On average, Reprompting still outperforms CoT by +12.4 point on these tasks.
> -	Exposition: The motivation for finding effective CoT recipes using Gibbs sampling is that an effective CoT recipe for solving a set of problems should be the one that can be applied to generating accurate solutions for all these problems, i.e. a common reasoning path. The distribution of such recipes is unknown, but we know how the solutions to these problems should be correlated. Thus, we can use Gibbs sampling to address the problem. Empirically, we found that the CoT solutions produced by Reprompting typically end up following a similar format and reasoning path, which verifies that by applying Gibbs sampling along with other approximations, we are able to discover common reasoning path for solving a set of problems.
> -	The wall-clock time for running Reprompting largely depends on the rate limit and delay of API calls to these LLMs. In our experiment, it takes 4-10 hours to run Reprompting on a single task.
> -	The motivation for decoupling initialization and sampling is based on the observation that some LLMs (e.g. GPT3.5) are better at generating more diverse reasoning paths than some others (e.g. text-davinci-003). Thus, using the former ones as the initialization model leads to a higher chance of discovering better CoT recipes for a task.
> -	The CoT prompts produced by Reprompting typically end up following a similar format and reasoning path. This is beneficial for finding a common, effective reasoning path for a specific type of problem. But for problems where the reasoning paths may vary greatly, it is best to first cluster the problems based on their type (e.g. by prompting LLMs about the category of the problem, whether they belong to dynamic programming, optimization, graph search or other problem types) or the clustering of their zero-shot solutions.
>
> [1] Parameter-efficient transfer learning for NLP. Houlsby et al., ICML 2019.
>
> [2] The power of scale for parameter-efficient prompt tuning. Lester et al., 2021.
>
> [3] P-tuning: Prompt tuning can be comparable to fine tuning across scales and tasks. Liu et al., ACL 2022.
>
> [4] Lora: Low-rank adaptation of large language models. Hu et al., 2021.
>
> [5] Improving Prompt Tuning with Learned Prompting Layers. Zhu et al., 2023.
>
> [6] Complexity-Based Prompting for Multi-Step Reasoning. Fu et al., 2022.

---

### Official Review · Reviewer_BA1Z · 2023-11-01

**Soundness:** 3 good
**Presentation:** 3 good
**Contribution:** 2 fair
**Rating:** 5
**Confidence:** 3

**Summary:**

This work introduces a simple chain-of-thought (CoT) prompt optimization strategy that bares similarity to evolutionary algorithms.  They show that their method outperforms baselines such as few-shot prompting, human written CoT prompts, CoT with self-consistency decoding, and an adaption of Auto-CoT to use labeled data.

**Strengths:**

* The paper is well written and easy to follow
* The algorithm is conceptually simple and easy to grasp and implement
* The algorithm can be executed without access to model parameters (e.g through a closed source API)

**Weaknesses:**

* This work is performing optimization to learn the CoT prompts, but does not compare against other prompt optimization methods,  prompt tuning, or fine-tuning.  The fact that optimized prompts beat human written CoT prompts is not surprising by default.  To evaluate the merit of the method one would need to compare against other schemes that use the training data to optimize the prompts at hand.
* There is no ablation study.  It is not clear which elements of the algorithm are most important for it's success.

I do not feel comfortable accepting this paper until there is evaluation against prompt optimization schemes at a similar compute budget.  Currently, by simply comparing against baselines it is too hard to gauge the contribution of this paper.

**Questions:**

Table 1: Perhaps add row depicting average performance for each method

Consider using \eqref to reference equations as it’s helpful to have parenthesis around the number of the equation you’re referencing for readability.

---

> ### Author Response · Authors · 2023-11-22
>
> We thank the reviewer for the valuable comments and insightful questions. Here are some responses to the weaknesses and questions:
>
> - Compared with prompt tuning and fine-tuning approaches, the number of labeled samples required by our approach is much less. Prompt tuning and parameter-efficient fine-tuning approaches require 100 to several thousand labeled samples for training and validation [1,2,3,4,5], which is not suitable for the few-shot setting that we focus on in this paper (e.g. on BigBenchHard, some tasks only contain a few dozen labeled samples apart from the 250 test samples). By contrast, our approach only requires 20 labeled samples, and thus can be tested on these few-shot benchmarks. However, we did add an additional baseline that optimizes the CoT prompt based on human-written solutions (so it’s human effort + optimization) [7]. The results are presented in Table 4 in the updated Appendix. On the three commonsense and arithmetic reasoning tasks, our method outperforms the baseline by +5 points on average.
> -	The fact that optimized prompts can beat human-written ones on reasoning tasks is not unsurprising. For instance, a recently proposed approach on optimizing/adapting CoT prompts does not outperform human-written CoT on average [6]. However, our method does outperform human-written CoT by large margins on a wide range of reasoning tasks (see Table 1 and the additional results on other tasks in Table 3 in Appendix).
> -	We further conducted an ablation study on the rejection sampling and recombination process. Results in Table 5 (in Appendix) show that, without rejection sampling, the test performance degrades greatly. Always rejecting solutions that lead to incorrect answers also leads to degradation. Additionally, not allowing multiple solutions to be recombined when sampling new solutions at the iterative sampling stage also hurts performance. Furthermore, since the Auto-CoT baseline in our experiment differs from our method in that it only has the initial sampling step without the iterative sampling step, it also serves as an ablative baseline. The comparison results show that the iterative sampling step brings +11 point improvement on average on the five BBH tasks.
>
> [1] Parameter-efficient transfer learning for NLP. Houlsby et al., ICML 2019.
>
> [2] The power of scale for parameter-efficient prompt tuning. Lester et al., 2021.
>
> [3] P-tuning: Prompt tuning can be comparable to fine tuning across scales and tasks. Liu et al., ACL 2022.
>
> [4] Lora: Low-rank adaptation of large language models. Hu et al., 2021.
>
> [5] Improving Prompt Tuning with Learned Prompting Layers. Zhu et al., 2023.
>
> [6] Better Zero-Shot Reasoning with Self-Adaptive Prompting, Wan et.al., ACL23.
>
> [7] Complexity-Based Prompting for Multi-Step Reasoning. Fu et al., 2022.

---

### Official Review · Reviewer_hjgQ · 2023-11-03

**Soundness:** 3 good
**Presentation:** 3 good
**Contribution:** 3 good
**Rating:** 6
**Confidence:** 3

**Summary:**

This paper proposes to automatically figure out the best K-shot chain-of-thoughts (CoT) examples that can be universally used in a task domain. The main idea is to leverage an iterative sampling method that is analogous to Gibbs sampling, where the algorithm starts from the ‘thoughts’ generated by the zero-shot CoT, and then in each iteration a random set of K-shot CoT is selected as the conditioning part, in order to regenerate the ‘thoughts’ for a given example. Experiments compared with several baselines show the consistent improvement, and the algorithm seems to be able to converge w.r.t t the training accuracy over the iterations.

**Strengths:**

- Automatically construction of the prompt is an important problem to tackle.
- The idea that is analogous to the Gibbs sampling seems to be novel and interesting.
- The empirical gain seems to be consistent, and the empirical justification of the algorithmic convergence seems to be convincing.

**Weaknesses:**

- Some more justification on the approximation of the equation 4) needs to be provided. Specifically, the eq 4) suggests some form of conditional independence structure in the fully connected graphical model, but how well it approximates (or how the neighborhood size matters) needs to be justified.

- The few-shot experiment setting might need a bit more justification. For example if the authors used 20 examples during CoT generation, then it might be tricky to call it a K-shot learning when K examples are used during the inference stage.


- Some latest baselines regarding the automatic prompt engineering need to be discussed or compared if appropriate, for example [1][2].


>**references**:\
[1] Better Zero-Shot Reasoning with Self-Adaptive Prompting, Wan et.al, ACL23 \
[2] Tempera: Test-time prompt editing via reinforcement learning, Zhang et.al, ICLR23

**Questions:**

I’d like to see the authors’ response regarding my concerns listed above, namely
- the quality w.r.t. the neighborhood size during Gibbs sampling
- the experiment setup regarding the k-shot learning
- the questions regarding the baselines

---

> ### Author Response · Authors · 2023-11-22
>
> We thank the reviewer for the valuable comments and insightful questions. Here are some responses to the weaknesses and questions:
>
> - In Eq(4), the first proportionality is deduced based on Bayes’ Theorem. The second approximation is based on the in-context learning assumption discussed in Eq(2). This assumption is based on the empirical findings in [3,4] on a wide range of tasks that the test performance remains similar when there is a varying number (but typically >= 5) of few-shot exemplars. We have updated Section 2.1 to better explain it.
> -	Our method uses 20 examples for CoT prompt generation, and the final prompt contains 5 example CoT. Since 20 labeled samples are used, it satisfies the definition of 20-shot learning. This is slightly different from the K-shot learning in standard few-shot prompting where K is the number of examples in the final prompt. But our comparison with few-shot prompting is still fair, since we are comparing our method with 20-shot prompting. As for the CoT baseline, since the number of CoT demonstrations is limited (and often less than 20) and each individual demonstration takes more tokens, we just take all the available CoT demonstrations that could fit into the prompt under the token limit.
> -	[1] proposed a novel self-adaptive prompting approach to improve model performance on reasoning tasks in zero-shot settings. In their approach, they generate step-by-step solutions through zero-shot CoT prompting and select demonstration examples from the zero-shot solutions via carefully designed criteria. This is different from our approach in that our approach is iterative so that zero-shot solutions can be recombined (Figure 4) and evolve into better recipes (Figure 3) through iterations. Empirically, since the approach in [1] does not outperform standard CoT (one of our baselines) on average, and their code has not been released yet, we didn’t include [1] as a baseline. But we will incorporate the discussion in paper. [2] proposed a novel prompt tuning method via reinforcement learning. The approach is based on a set of prompt editing operations that are designed for text classification tasks. Such approach is not directly applicable to reasoning tasks where the space of all possible step-by-step reasoning paths is huge, and one needs to explore the space by discovering and recombining new paths for solving the problem rather than simply rephrasing existing ones. We will incorporate the discussion and citations in the paper.
>
> [3] Language Models are Few-Shot Learners. Brown et al., NeurIPS 2020.
>
> [4] Chain-of-thought prompting elicits reasoning in large language models. Wei et al., NeurIPS 2022.

---

### Author Response · Authors · 2023-11-22

We thank all reviewers for the valuable comments and insightful questions. We have incorporated the suggestions in the updated version of the paper. We have also evaluated our method on 10 additional tasks/benchmarks based on all reviewers’ suggestions. Due to the page limit, the additional experimental results are updated to the Appendix. We will further incorporate these additional results in the main paper given more space.

---

### Meta-Review · Area_Chair_FHum · 2023-12-05

**Metareview:**

This paper proposes reprompting which aims to automatically generate k-shot chain-of-thought examples that can be universally used across a task domain. This is achieved by a Gibbs-sampling procedure that infers CoT prompts that work well for a set of training examples.

I recommend rejecting this paper for the following reasons:
- The paper seems to have strong assumptions.
- The paper does not include comparisons against other prompt optimization methods, prompt tuning and finetuning.

**Justification For Why Not Higher Score:**

I recommend rejecting this paper for the following reasons:
- The paper seems to have strong assumptions.
- The paper does not include comparisons against other prompt optimization methods, prompt tuning and finetuning.

**Justification For Why Not Lower Score:**

N/A

---

### Decision · Program_Chairs · 2024-01-16

Reject